# Towards Effective Surgical Representation Learning with DINO Models

**Ronald L.P.D. de Jong**[1] (ID)                     R.L.P.D.D.JONG@TUE.NL
**Yiping Li**[1] (ID)                                 Y.LI9@TUE.NL
**Tim J.M. Jaspers**[2] (ID)                          T.J.M.JASPERS@TUE.NL
**Romy C. van Jaarsveld**[3] (ID)                     R.C.VANJAARSVELD-3@UMCUTRECHT.NL
**Gino M. Kuiper**[3] (ID)                            G.M.KUIPER-2@UMCUTRECHT.NL
**Franco Badaloni**[3] (ID)                           F.BADALONI@UMCUTRECHT.NL
**Richard van Hillegersberg**[3] (ID)                 R.VANHILLEGERSBERG@UMCUTRECHT.NL
**Jelle P. Ruurda**[3] (ID)                           J.P.RUURDA@UMCUTRECHT.NL
**Fons van der Sommen**[2] (ID)                       FVDSOMMEN@TUE.NL
**Josien P.W. Pluim**[1] (ID)                         J.PLUIM@TUE.NL
**Marcel Breeuwer**[1,2] (ID)                         M.BREEUWER@TUE.NL

[1] *Department of Biomedical Engineering, Eindhoven University of Technology, Groene Loper 3, Eindhoven, 5612 AE, Noord-Brabant, The Netherlands.*

[2] *Department of Electrical Engineering, Eindhoven University of Technology, Groene Loper 3, Eindhoven, 5612 AE, Noord-Brabant, The Netherlands.*

[3] *Department of Surgery, University Medical Center Utrecht, Heidelberglaan 100, Utrecht, 3584 CX, Utrecht, The Netherlands.*

**Editors:** Accepted for publication at MIDL 2026

## Abstract

Self-supervised learning (SSL) has emerged as a promising approach to address the limitations of annotated surgical datasets, which are often small, heterogeneous, and expensive to curate. Among SSL methods, self-distillation with no labels (DINO) has achieved state-of-the-art (SOTA) results in natural images, but its applicability to surgical data remains underexplored. In this work, we systematically investigate DINOv1, DINOv2, and DINOv3 for surgical representation learning. We pretrain these models on a large-scale surgical dataset of 4.7M video frames (SurgeNetXL) and evaluate their transferability on downstream tasks including semantic segmentation and surgical phase recognition. Our results demonstrate that in-domain pretraining consistently improves performance across all DINO variants, with DINOv2 and DINOv3 achieving SOTA performance. We further offer practical insights and visualizations highlighting the effectiveness of SSL. Finally, our study delivers ready-to-use DINO-based SSL models and pretraining protocols for surgical computer vision research, which are publicly available at: github.com/rlpddejong/SurgeNetDINO.

**Keywords:** Anatomy recognition, DINO, Foundation models, Self-supervised learning, Surgical phase recognition

## 1. Introduction

The success of deep learning in surgical data science has largely relied on supervised models trained on curated datasets. However, collecting and annotating surgical data is challenging: annotations require expert knowledge, surgical procedures are highly variable across centers,

and available datasets are often small compared to natural image benchmarks. These constraints limit the scalability of purely supervised approaches and motivate the use of self-supervised learning (SSL), where models are first pretrained on large amounts of unlabeled surgical data and then fine-tuned in a supervised manner on diverse downstream surgical tasks.

Self-distillation with no labels (DINO) has emerged as a leading SSL method in natural images, with successive versions (DINOv1, DINOv2, DINOv3) producing increasingly robust visual representations (Caron et al., 2021; Oquab et al., 2024; Siméoni et al., 2025). Despite their success in general computer vision, the utility of these methods for surgical applications remains largely unexplored. In particular, it is unclear how the different DINO variants compare to each other on surgical data science tasks, to what extent surgical pretraining improves their performance on downstream tasks, and which training strategies are most effective in this domain.

In this work, we investigate self-supervised pretraining of DINOv1, v2, and v3 on a large-scale surgical dataset of 4.7M frames, followed by supervised fine-tuning on a variety of downstream surgical computer vision tasks. This two-stage approach allows us to evaluate the transferability of pretrained representations and identify the design choices that maximize performance on each task. Beyond reporting comparative results, we examine key training parameters to offer practical recommendations for adapting SSL to surgical data. Our contributions can be summarized as follows:

- We present the first systematic study of DINOv1, v2, and v3 pretrained on a large-scale surgical dataset consisting of 4.7M frames, and release the model weights at: github.com/rlpddejong/SurgeNetDINO.

- We provide comparative insights into the transferability and strengths of each DINO variant for various downstream surgical computer vision applications, achieving state-of-the-art (SOTA) performance that surpasses prior methods.

- We perform analyses on architecture size, training duration, and fine-tuning strategies, providing practical recommendations for designing effective SSL pipelines in the surgical domain.

## 2. Related work

### 2.1. Self-supervised learning (SSL)

SSL has emerged as a powerful paradigm for learning useful representations from unlabeled data, reducing reliance on costly manual annotations. Early approaches in SSL relied on pretext tasks such as image inpainting, colorization, or predicting rotations (Pathak et al., 2016; Larsson et al., 2016; Gidaris et al., 2018). More recent contrastive learning methods, such as SimCLR (Chen et al., 2020) and MoCo (He et al., 2020), learn representations by encouraging similarity between augmented views of the same image while pushing apart representations of different images. These methods have shown remarkable success on natural image benchmarks.

## 2.2. Self-distillation with no labels (DINO)

DINO is an SSL method that avoids explicit contrastive objectives by leveraging knowledge distillation between student and teacher networks (Caron et al., 2021). Subsequent versions, DINOv2 and DINOv3, have improved representation quality and generalization by employing stronger training protocols, larger datasets, and better architectural choices (Oquab et al., 2024; Siméoni et al., 2025). DINO has demonstrated SOTA performance in a variety of downstream tasks in natural image domains, including classification, detection, and segmentation. However, most evaluations of these methods, especially the latest versions, focus primarily on general computer vision datasets, such as ImageNet.

## 2.3. SSL in surgical computer vision

Recent work has examined the potential of DINO in the surgical computer vision domain. Ramesh et al. (2023) systematically analyzed self-supervised methods, including DINO, for surgical computer vision tasks and highlighted their ability to improve representation learning. Cui et al. (2024) proposed Surgical-DINO, adapting foundation models with adapters for depth estimation in endoscopic surgery. More recently, Jaspers et al. (2026) showed that scaling self-supervised pretraining on large surgical datasets can substantially enhance the performance of foundation models for a variety of surgical computer vision tasks. Additionally, the LEMON dataset (Che et al., 2025) highlights the growing importance of large-scale data curation efforts in this field, with a particular focus on organizing and refining extensive unlabeled endoscopic video collections for foundation model training. Despite these advances, important gaps remain: DINOv3 has not yet been explored in surgical applications, and no direct comparisons between different DINO variants have been performed. These limitations leave open questions about the most effective pretraining strategies and model choices for this specialized domain.

## 3. Methods

### 3.1. Pretraining datasets

The original DINO models were pretrained on large-scale natural image datasets, such as ImageNet (Russakovsky et al., 2014), LVD-142M (Oquab et al., 2024), and LVD-1689M (Siméoni et al., 2025), which provide strong visual priors for general computer vision tasks. However, these datasets are limited in their relevance to surgical domains, as they lack the anatomical structures, tools, and visual characteristics present in operative scenes.

For self-supervised pretraining, we used the SurgeNetXL dataset introduced by Jaspers et al. (2026). SurgeNetXL is a large-scale collection of 4.7M surgical video frames sampled at 1 fps from a combination of public surgical datasets, curated YouTube surgical footage, and two institutional datasets. All videos were anonymized where required, and quality-filtered to exclude non-surgical or irrelevant content. By integrating diverse procedures and video sources, SurgeNetXL provides both scale and heterogeneity, making it well-suited for pretraining large vision models in surgical domains.

## 3.2. Pretraining implementation

We pretrained DINOv1, DINOv2, and DINOv3 following their original frameworks on the small, base, and large variants of the Vision Transformer (ViT-S, ViT-B, and ViT-L, respectively), models originally proposed by Dosovitskiy et al. (2021). Notably, DINOv1 was not pretrained on ViT-L, as this variant was not included in the original implementation. The trained model weights, along with the exact pretraining details, are publicly available at github.com/rlpddejong/SurgeNetDINO. Pretraining was initialized from ImageNet, LVD-142M, and LVD-1689M weights for DINOv1, DINOv2, and DINOv3, respectively. All pretraining experiments were carried out on four NVIDIA H100 GPUs (NVIDIA Corp., CA, USA) using the largest feasible batch size for each model during 120 hours, with a maximum of 50 epochs.

## 3.3. Fine-tuning datasets

In this study, we evaluated our pretrained models on four downstream datasets covering two key tasks: semantic segmentation and surgical phase recognition. These datasets include a range of surgical procedures such as laparoscopic cholecystectomy, robot-assisted minimally invasive esophagectomy (RAMIE), and hysterectomy. For semantic segmentation, we use the CholecSeg8k dataset (Hong et al., 2020), and the RAMIE-seg dataset (de Jong et al., 2025), which contain pixel-level annotations of anatomical structures and surgical tools. These datasets represent both laparoscopic and robotic procedures, with varying numbers of patients, frames, and classes. Surgical phase recognition was performed on the Auto-Laparo dataset (Wang et al., 2022) and an extended version of the RAMIE-phase dataset proposed by Li et al. (2025). These datasets consist of full-length surgical recordings annotated with temporal phases, including seven phases for hysterectomy and thirteen phases for esophagectomy. The combination of these datasets offers a diverse and representative benchmark for evaluating the effectiveness of self-supervised pretraining in multiple clinically relevant downstream tasks, supporting potential applications such as real-time surgical guidance, workflow monitoring, and postoperative analysis.

## 3.4. Fine-tuning implementation

### 3.4.1. SEMANTIC SEGMENTATION

For semantic segmentation, we adopted the recently introduced Encoder-only Mask Transformer (EoMT) (Kerssies et al., 2025). EoMT uses a ViT backbone with learnable object queries and a lightweight mask–class head, avoiding the need for task-specific decoders. We used EoMT because it enables efficient fine-tuning on small surgical datasets, reduces architectural complexity, and aligns well with our goal of evaluating the quality of DINO-based representations without confounding effects from heavy decoder designs.

For CholecSeg8k, we used 6,800 frames for training and 1,280 frames for testing. Consistent with previous studies (Grammatikopoulou et al., 2024; Zhang et al., 2024), we excluded low-prevalence classes (blood, cystic duct, hepatic vein, and liver ligament) to ensure a robust evaluation. For RAMIE, 749 frames were allocated for training and 120 frames for testing. All splits were performed on a per-patient basis.

The model was trained using five-fold cross-validation at the patient level, with approximately 80% of the data used for training and 20% for validation. All frames were resized to match the resolution used during pretraining, using bicubic interpolation. We used the loss function described by Kerssies et al. (2025) and the AdamW optimizer with a learning rate of $1\times10^{-5}$, which was reduced by half after 10 epochs without an improvement in validation loss. All models were trained on a single NVIDIA H100 GPU (NVIDIA Corp., CA, USA) with a batch size of 8 and an early-stopping criterion of 15 epochs. Data augmentation was limited to horizontal and vertical flips and rotations, each applied with a 50% probability.

### 3.4.2. Phase recognition

For fine-tuning on the phase recognition datasets, we adopted a two-stage training procedure following Jaspers et al. (2026). In the first stage, the backbone model was trained to predict surgical phases from individual frames. In the second stage, a causal Multi-Stage Temporal Convolutional Network (MS-TCN) was trained on the extracted features.

For the AutoLaparo dataset (Wang et al., 2022), we followed the original train, validation, and test splits. For the extended RAMIE-phase dataset, we used 18 videos for training, 7 for validation, and 16 for testing. Frames were extracted at 1 fps and resized using bicubic interpolation to match the pretraining resolution of each backbone.

Both training stages used the AdamW optimizer and cross-entropy loss. In the first stage, the backbone was trained on individual frames with a learning rate of $1 \times 10^{-5}$ and the batch size was 32 for ViT-L and 64 for ViT-S and ViT-B. In the second stage, the causal MS-TCN was trained on full-video feature sequences with a batch size of 1, a learning rate of $7 \times 10^{-4}$, and 200 epochs. Data augmentation was applied only in the first stage and included random scaling and rotation, RGBShift, and RandomBrightnessContrast. All models were trained on a single NVIDIA A100 GPU (NVIDIA Corp., CA, USA).

## 3.5. Evaluation

For semantic segmentation, model performance was assessed using the Dice score, which measures the overlap between predicted and annotated masks, and the 95th percentile Hausdorff distance (HD95), which quantifies boundary alignment. Metrics were first averaged across classes for each patient, and then across patients to obtain overall scores. For surgical phase recognition, we report both frame-wise accuracy and macro-averaged F1 score, meaning that the F1 score is computed per phase and then averaged across phases. Reported results are expressed as mean $\pm$ standard deviation. To benchmark our models, we fine-tuned the following SOTA models: CAFormer (Jaspers et al., 2026), EndoFM (Wang et al., 2023), EndoViT (Batić et al., 2023), GSViT (Schmidgall et al., 2024), SAM2-UNet (Xiong et al., 2026), and SegFormer (Xie et al., 2021). To ensure a fair comparison, all SOTA models were fine-tuned using the same training configuration. For GSViT, we implemented an FPN decoder, as the original model did not include a decoder for semantic segmentation. We use a bootstrapping approach (Wiesenfarth et al., 2021; Varoquaux and Cheplygina, 2022) to evaluate the stability of our models with the SOTA models. The predictions are pooled to generate 1,000 bootstrap samples with replacement, maintaining a consistent random seed across experiments. This procedure accounts for sampling variability, which is especially important for small datasets or limited patient numbers.

## 4. Results

Table 1 presents the performance of DINOv1–v3 variants with various ViT architectures and pretraining strategies on the CholecSeg8k and RAMIE-seg datasets. Pretrained models consistently outperform those trained from scratch. In particular, SurgeNetXL in-domain pretraining improves performance across all DINO variants compared to ImageNet or LVD, with an average improvement of 9.1%. It is worth noting that DINOv2 and DINOv3 models pretrained on the LVD datasets achieve performance comparable to DINOv1 models pretrained on SurgeNetXL, highlighting the advancements introduced in the newer versions of DINO. Larger backbones in DINOv2 and DINOv3 (ViT-B and ViT-L) further enhance segmentation performance. Specifically, DINOv2 with a ViT-L backbone reaches Dice scores of up to 0.77 and 0.79 on CholecSeg8k and RAMIE-seg, respectively, with substantially reduced HD95. DINOv3 reinforces these trends, with the ViT-L model achieving the highest Dice scores (0.78 and 0.74) and the lowest HD95 (29 and 46) compared to ViT-S and ViT-B. These results demonstrate that both DINOv2 and DINOv3 are strong segmentation models, particularly when paired with larger backbones and domain-specific pretraining.

Table 1: Performance comparison of DINOv1–v3 variants with different ViT architectures and pretraining strategies on the CholecSeg8k and RAMIE-seg datasets. Vanilla ViT architectures trained from scratch are included as baselines.

| Model | | Pretraining | CholecSeg8k dataset | | RAMIE-seg dataset | |
|---|---|---|---|---|---|---|
| | | | Dice score ↑ | HD95 ↓ | Dice score ↑ | HD95 ↓ |
| | ViT-S | | $0.52 \pm 0.05$ | $100 \pm 16$ | $0.35 \pm 0.04$ | $133 \pm 16$ |
| | ViT-B | No pretraining | $0.52 \pm 0.07$ | $101 \pm 16$ | $0.41 \pm 0.06$ | $125 \pm 9$ |
| | ViT-L | | $0.55 \pm 0.10$ | $102 \pm 16$ | $0.44 \pm 0.05$ | $123 \pm 6$ |
| v1 | ViT-S | ImageNet | $0.66 \pm 0.09$ | $65 \pm 25$ | $0.57 \pm 0.04$ | $73 \pm 11$ |
| | | SurgeNetXL | $\mathbf{0.73 \pm 0.09}$ | $\mathbf{56 \pm 19}$ | $0.62 \pm 0.03$ | $63 \pm 6$ |
| | ViT-B | ImageNet | $0.68 \pm 0.07$ | $61 \pm 23$ | $0.61 \pm 0.02$ | $69 \pm 9$ |
| | | SurgeNetXL | $\mathbf{0.73 \pm 0.10}$ | $57 \pm 19$ | $\mathbf{0.67 \pm 0.03}$ | $\mathbf{58 \pm 14}$ |
| v2 | ViT-S | LVD-142M | $0.70 \pm 0.07$ | $54 \pm 19$ | $0.60 \pm 0.04$ | $70 \pm 7$ |
| | | SurgeNetXL | $0.66 \pm 0.06$ | $55 \pm 18$ | $0.59 \pm 0.03$ | $67 \pm 5$ |
| | ViT-B | LVD-142M | $0.71 \pm 0.06$ | $50 \pm 18$ | $0.67 \pm 0.05$ | $52 \pm 6$ |
| | | SurgeNetXL | $0.75 \pm 0.10$ | $44 \pm 22$ | $0.73 \pm 0.04$ | $46 \pm 11$ |
| | ViT-L | LVD-142M | $0.63 \pm 0.09$ | $58 \pm 13$ | $0.70 \pm 0.05$ | $49 \pm 6$ |
| | | SurgeNetXL | $\mathbf{0.77 \pm 0.09}$ | $\mathbf{38 \pm 20}$ | $\mathbf{0.79 \pm 0.04}$ | $\mathbf{40 \pm 7}$ |
| v3 | ViT-S | LVD-1689M | $0.73 \pm 0.06$ | $52 \pm 21$ | $0.63 \pm 0.06$ | $59 \pm 6$ |
| | | SurgeNetXL | $0.74 \pm 0.07$ | $48 \pm 17$ | $0.69 \pm 0.05$ | $54 \pm 8$ |
| | ViT-B | LVD-1689M | $0.71 \pm 0.07$ | $55 \pm 24$ | $0.67 \pm 0.04$ | $53 \pm 10$ |
| | | SurgeNetXL | $0.75 \pm 0.09$ | $42 \pm 22$ | $0.73 \pm 0.05$ | $55 \pm 14$ |
| | ViT-L | LVD-1689M | $0.76 \pm 0.09$ | $34 \pm 21$ | $0.70 \pm 0.04$ | $\mathbf{46 \pm 3}$ |
| | | SurgeNetXL | $\mathbf{0.78 \pm 0.11}$ | $\mathbf{29 \pm 15}$ | $\mathbf{0.74 \pm 0.04}$ | $46 \pm 12$ |

Table 2 presents the performance of DINOv1–v3 variants with various ViT architectures and pretraining strategies on the phase recognition datasets. Similar to segmentation, pretraining consistently improves performance compared to training from scratch. Additionally, in-domain pretraining outperforms general pretraining, with a few exceptions for certain model variants. Interestingly, for phase recognition, the benefit of larger architectures is less pronounced, with DINOv1 even performing slightly better on ViT-S. Furthermore, although DINOv2 and DINOv3 achieve the highest scores, the differences between versions are smaller compared to segmentation.

Table 2: Performance comparison of DINOv1–v3 variants with different ViT architectures and pretraining strategies on the AutoLaparo and RAMIE-phase datasets. Vanilla ViT architectures trained from scratch are included as baselines.

| Model | | Pretraining | AutoLaparo dataset | | RAMIE-phase dataset | |
|---|---|---|---|---|---|---|
| | | | Accuracy ↑ | F1 score ↑ | Accuracy ↑ | F1 score ↑ |
| | ViT-S | | $53.9 \pm 14.8$ | $45.8 \pm 16.8$ | $63.1 \pm 14.4$ | $52.1 \pm 16.0$ |
| | ViT-B | No pretraining | $67.9 \pm 18.3$ | $59.5 \pm 16.5$ | $64.2 \pm 12.6$ | $51.0 \pm 14.3$ |
| | ViT-L | | $65.6 \pm 15.5$ | $58.1 \pm 13.5$ | $64.3 \pm 14.5$ | $52.3 \pm 16.2$ |
| v1 | ViT-S | ImageNet | $81.8 \pm 9.4$ | $70.4 \pm 6.8$ | $74.1 \pm 9.3$ | $65.3 \pm 10.0$ |
| | | SurgeNetXL | $\mathbf{85.3 \pm 5.6}$ | $\mathbf{75.1 \pm 5.2}$ | $\mathbf{77.4 \pm 9.8}$ | $\mathbf{70.5 \pm 11.8}$ |
| | ViT-B | ImageNet | $81.1 \pm 9.0$ | $68.7 \pm 8.4$ | $75.4 \pm 10.0$ | $66.4 \pm 10.9$ |
| | | SurgeNetXL | $85.0 \pm 9.1$ | $70.6 \pm 7.6$ | $77.2 \pm 10.6$ | $70.1 \pm 11.5$ |
| v2 | ViT-S | LVD-142M | $83.0 \pm 9.9$ | $73.0 \pm 11.0$ | $76.6 \pm 9.5$ | $67.9 \pm 9.9$ |
| | | SurgeNetXL | $84.5 \pm 8.6$ | $69.6 \pm 6.4$ | $76.0 \pm 11.0$ | $66.3 \pm 12.2$ |
| | ViT-B | LVD-142M | $85.3 \pm 7.4$ | $74.2 \pm 8.2$ | $77.4 \pm 9.9$ | $67.8 \pm 10.4$ |
| | | SurgeNetXL | $85.9 \pm 8.3$ | $74.0 \pm 5.9$ | $78.2 \pm 10.2$ | $71.1 \pm 11.8$ |
| | ViT-L | LVD-142M | $\mathbf{86.1 \pm 7.6}$ | $\mathbf{77.8 \pm 8.9}$ | $77.9 \pm 10.3$ | $70.2 \pm 11.4$ |
| | | SurgeNetXL | $84.1 \pm 8.4$ | $70.5 \pm 8.0$ | $\mathbf{80.8 \pm 9.1}$ | $\mathbf{73.7 \pm 11.5}$ |
| v3 | ViT-S | LVD-1689M | $81.1 \pm 10.0$ | $69.7 \pm 8.6$ | $75.4 \pm 10.8$ | $63.3 \pm 11.8$ |
| | | SurgeNetXL | $82.2 \pm 8.5$ | $72.1 \pm 8.0$ | $74.4 \pm 10.9$ | $62.9 \pm 11.7$ |
| | ViT-B | LVD-1689M | $83.4 \pm 9.4$ | $70.8 \pm 10.1$ | $75.8 \pm 10.7$ | $66.8 \pm 10.5$ |
| | | SurgeNetXL | $83.3 \pm 8.5$ | $73.0 \pm 4.7$ | $76.3 \pm 10.5$ | $66.7 \pm 12.1$ |
| | ViT-L | LVD-1689M | $86.0 \pm 8.6$ | $75.0 \pm 9.6$ | $77.8 \pm 9.2$ | $69.8 \pm 10.3$ |
| | | SurgeNetXL | $\mathbf{86.4 \pm 6.6}$ | $\mathbf{78.5 \pm 6.7}$ | $\mathbf{78.0 \pm 10.5}$ | $\mathbf{70.1 \pm 11.8}$ |

Figure 1 illustrates the ranking stability of our best models on segmentation and phase recognition tasks in comparison with top-performing SOTA models reported in literature. From the figure, it is evident that our models demonstrate the best overall performance, outperforming both SSL- and non-SSL-based approaches. This highlights the effectiveness of DINO pretraining in enhancing model robustness and stability across different evaluation scenarios.

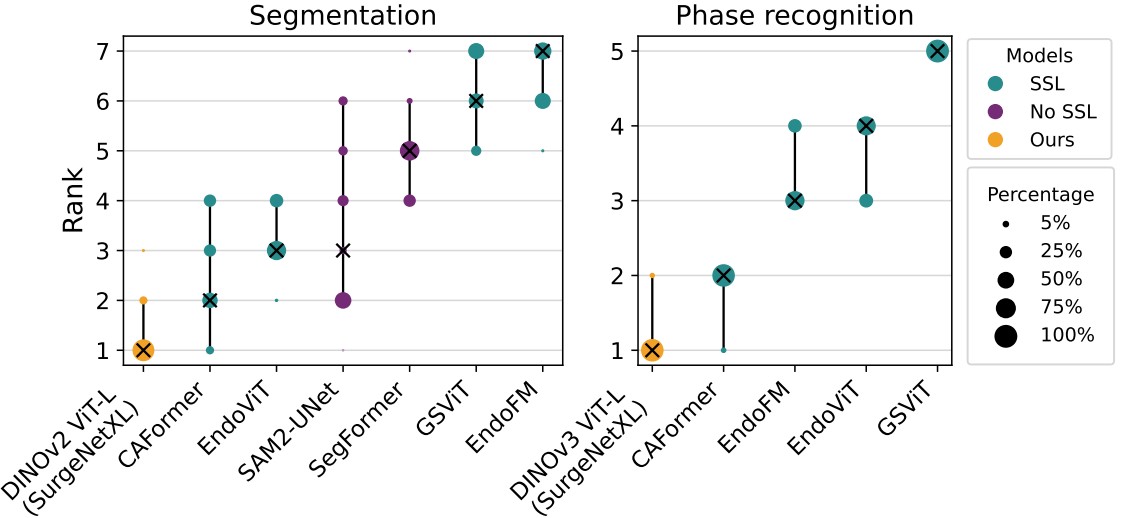

Figure 1: Ranking stability of DINO models vs. SSL and non-SSL pretrained SOTA models: CAFormer (Jaspers et al., 2026), EndoFM (Wang et al., 2023), EndoViT (Batić et al., 2023), GSViT (Schmidgall et al., 2024), SAM2-UNet (Xiong et al., 2026), and SegFormer (Xie et al., 2021). Results are based on all metrics and datasets in Table 1 and Table 2, respectively. Blob size indicates how often an architecture attains a given rank. Black crosses mark the median ranks rounded to nearest integer, and black lines show 95% bootstrap intervals (2.5–97.5th percentiles). Models are ordered left to right by mean bootstrap rank, best to worst.

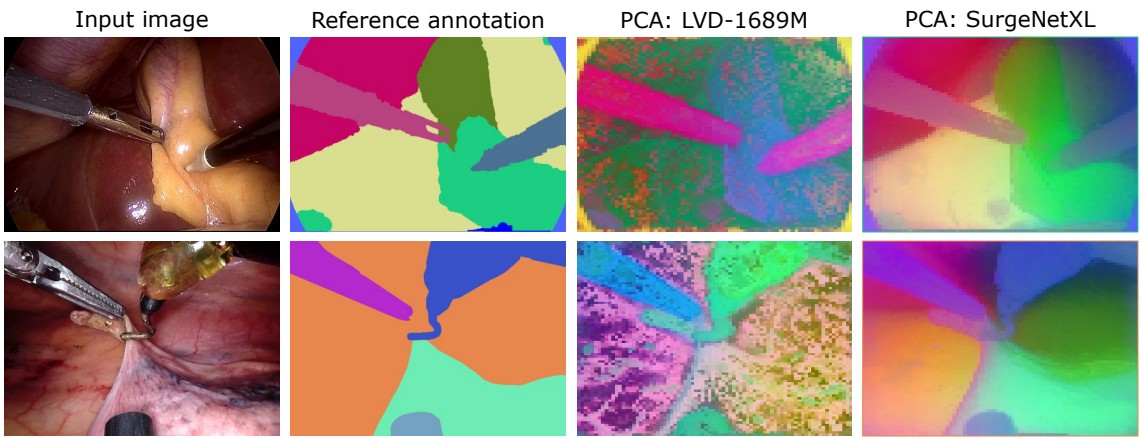

Figure 2: PCA of unsupervised DINOv2 ViT-L features on CholecSeg8k (top row) and RAMIE-seg (bottom row). Columns show: input image, reference annotation, PCA of LVD-1689M features, and PCA of SurgeNetXL features.

Figure 2 presents a principal component analysis (PCA) of features extracted by the unsupervised DINOv2 ViT-L model. For visualization, the three principal components were mapped to the RGB color channels. To better resemble the similarity of the PCA to the reference annotations, the colors of the annotations have been manually matched retrospectively. The PCA reveals that models pretrained on SurgeNetXL better capture the semantic meaning of the surgical images: most annotated structures are represented as distinct, coherent colors, which is remarkable given that the model was trained entirely with self-supervision. In contrast, the LVD-1689M features produce noisier results, with most structures less clearly separated.

The left plot in Figure 3 shows the Dice score improvements for each of the DINO variants over training time. All models improve as training progresses, highlighting the importance of pretraining duration. Interestingly, DINOv3 exhibits slightly lower final score improvements, which may be due to its stronger pretrained features compared to the other variants, reducing the apparent gain from additional training. A similar, albeit smaller, effect is also observed for DINOv1 and DINOv2, which could indicate slight overfitting.

The right plot in Figure 3 shows the Dice scores for DINO models pretrained on ImageNet, LVD variants, and SurgeNetXL, comparing frozen and trainable encoders during fine-tuning. While median scores are slightly higher with a trainable encoder, the differences are minimal, and using a frozen encoder significantly reduces training complexity due to fewer trainable parameters. This also demonstrates the high quality of DINO features, for which extensive fine-tuning is not necessary.

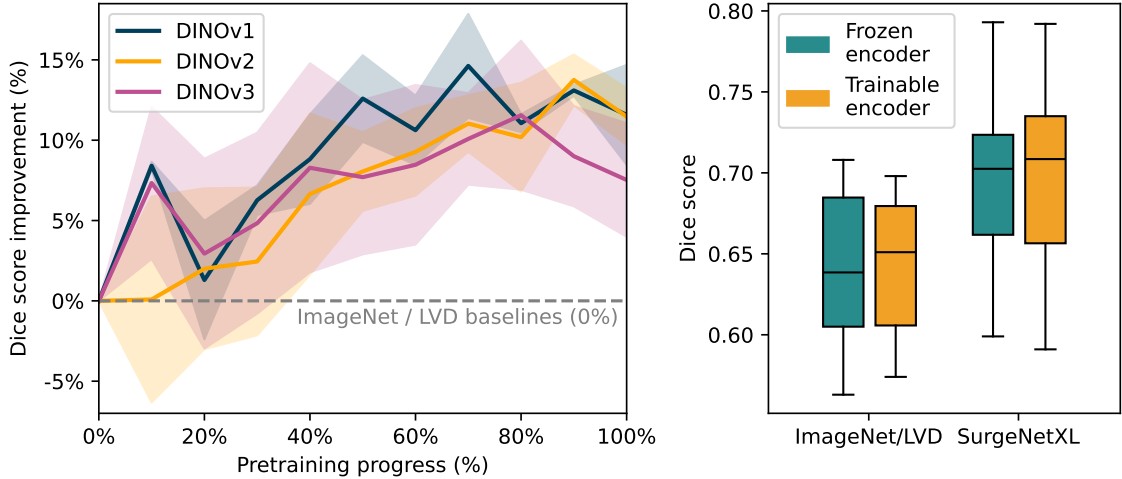

Figure 3: Left: Dice score improvements of DINOv1–v3 relative to the ImageNet and LVD baselines over training time. Lines represent the mean, and shaded areas indicate the standard deviation. Right: Boxplots showing the Dice scores for all models based on a frozen or trainable encoder during fine-tuning.

Table 3 reports online inference efficiency of DINO variants and SOTA models for segmentation and phase recognition. Note that the number of parameters between segmenta-

tion and phase recognition models may differ due to the different heads used for each task. Smaller ViT variants achieve the lowest latency and highest FPS, with DINOv2 ViT-S reaching 227 FPS on segmentation, making them suitable for real-time surgical applications. Larger models offer higher downstream accuracy but exhibit lower FPS (31–35 for phase recognition), indicating a trade-off between performance and speed. Compared to existing SOTA models, DINO variants provide competitive or superior throughput while maintaining strong task performance, demonstrating their practical applicability for real-time surgical video analysis.

Table 3: Online inference efficiency comparison for DINOv1–v3 variants and SOTA models. Params denotes the total number of model parameters (in millions, M), latency is measured in milliseconds (ms), and FPS denotes the frames per second. All experiments were caried out on a single NVIDIA H100 GPU.

| Model | | Segmentation | | | Phase recognition | | |
|---|---|---|---|---|---|---|---|
| | | Params (M) | Latency (ms) | FPS | Params (M) | Latency (ms) | FPS |
| CAFormer | | 25 | 5.3 | 189 | 25 | 15.3 | 65 |
| EndoFM | | 141 | 12.7 | 79 | 122 | 20.5 | 49 |
| EndoViT | | 111 | 7.4 | 134 | 87 | 11.2 | 89 |
| GSViT | | 340 | 8.3 | 120 | 12 | 22.0 | 45 |
| SAM2-UNet | | 4 | 13.3 | 76 | - | - | - |
| SegFormer | | 4 | 3.5 | 286 | - | - | - |
| v1 | ViT-S | 23 | 4.5 | 222 | 22 | 9.6 | 104 |
| | ViT-B | 92 | 4.7 | 213 | 86 | 11.3 | 89 |
| v2 | ViT-S | 23 | 4.4 | 227 | 22 | 10.8 | 92 |
| | ViT-B | 90 | 4.5 | 224 | 86 | 13.3 | 75 |
| | ViT-L | 311 | 8.1 | 123 | 304 | 28.9 | 35 |
| v3 | ViT-S | 23 | 6.5 | 153 | 22 | 14.4 | 69 |
| | ViT-B | 92 | 6.8 | 147 | 86 | 16.7 | 60 |
| | ViT-L | 314 | 10.8 | 93 | 304 | 32.6 | 31 |

Table 4 summarizes the computational requirements for pretraining DINOv3 on four NVIDIA H100 GPUs using the full SurgeNetXL dataset of approximately 4.7M images. We report the maximum batch size per GPU that fits in 94 GiB of memory, the resulting total batch size across all GPUs, the time per epoch, and the effective throughput in images per second. As expected, larger ViT backbones require smaller per-GPU batch sizes and longer epoch times, with ViT-L achieving the lowest throughput due to its higher memory and compute demands. Interestingly, ViT-S and ViT-B achieve similar throughput, likely because data loading becomes the primary bottleneck rather than model size. The reported time per epoch is based on a dataset of 4.7M images and would scale proportionally with smaller datasets, meaning that pretraining on subsets of surgical data would be substantially faster and more practical for researchers with limited computational resources.

Table 4: Computational cost of DINOv3 pretraining on 4 NVIDIA H100 GPUs. For each model size, we use the maximum feasible batch size that fits into GPU memory (94 GiB per GPU).

| Model | Batch size / GPU | Total batch size | Time / epoch (hours) | Images / second |
|-------|------------------|------------------|----------------------|-----------------|
| ViT-S | 192 | 768 | 2.19 | 598 |
| ViT-B | 128 | 512 | 2.19 | 598 |
| ViT-L | 64 | 256 | 3.36 | 389 |

## 5. Discussion

Our experiments provide several insights into the application of DINO models for surgical representation learning. First, in-domain pretraining on SurgeNetXL consistently improves performance across both semantic segmentation and surgical phase recognition tasks. This highlights the importance of domain-specific visual priors, which better capture the unique appearance of surgical scenes.

The improvements observed for segmentation and phase recognition differ in magnitude and behavior. Segmentation tasks benefit substantially from larger ViT backbones (ViT-B and ViT-L), as they require fine-grained spatial understanding and precise delineation of anatomical structures and surgical instruments. In contrast, phase recognition shows smaller gains from increased model capacity; even smaller ViT-S models achieve competitive accuracy. This difference likely reflects the inherent nature of the tasks: segmentation demands high-resolution spatial reasoning, whereas phase recognition relies primarily on temporal patterns and higher-level contextual cues, which smaller architectures seem to capture sufficiently.

Visualization of unsupervised features via PCA further supports our findings. Models pretrained on SurgeNetXL produce coherent feature clusters corresponding to anatomical structures, whereas models pretrained on natural images alone produce noisier and less semantically meaningful representations. This suggests that DINO can capture relevant surgical semantics without supervision, provided the pretraining data is representative of the domain.

Generally, DINOv2 and DINOv3 outperform DINOv1, with only minor differences between DINOv2 and DINOv3. One possible explanation is that for DINOv3 we only applied high-resolution fine-tuning. Further improvements could be achieved by training even larger ViT models, such as a ViT-7B, followed by knowledge distillation to smaller models like in the original implementation. However, it would require substantially more GPU resources to be effective.

Inference experiments indicate that all DINO variants comfortably exceed real-time requirements for segmentation, with the largest models achieving over 90 FPS and smaller variants exceeding 200 FPS. Although phase recognition models are slightly slower, they still meet real-time constraints, as phase information at around 30 FPS or lower is sufficient for most practical applications. Nevertheless, a clear trade-off remains between smaller and larger backbones, which becomes particularly relevant when deploying these models on resource-constrained or edge devices.

Our analysis shows that in-domain pretraining yields modest metric gains at a substantial added computational cost. However, these improvements are consistent across datasets, especially on the smaller segmentation sets, and represent a one-time investment whose benefits carry over to multiple downstream tasks, reducing annotation effort and boosting reliability. In surgical contexts, such transferable gains, particularly for rare structures or limited-data scenarios, can meaningfully enhance both performance and patient safety, justifying the computational cost of in-domain pretraining.

In addition, DINO-based pretraining on domain-specific surgical videos outperforms existing SOTA methods such as SAM2-UNet, even when those models are pretrained on large general video datasets. While foundation models like SAM 2 enable fast transfer via lightweight fine-tuning, they lack the specialized visual priors needed to capture the fine-grained anatomical and procedural details present in surgical scenes. By contrast, DINO models pretrained on SurgeNetXL learn domain-relevant representations that better encode surgical structures and instrument interactions, resulting in superior segmentation and phase recognition performance. These results indicate that careful in-domain self-supervised pretraining can provide measurable advantages over off-the-shelf foundation models, particularly in tasks requiring high spatial precision and clinical reliability.

We acknowledge that comparing DINO pretrained on SurgeNetXL to models pretrained on ImageNet or general video datasets involves a domain mismatch, so some performance gap is expected. Nevertheless, this comparison is informative, as it quantifies the benefits of domain-specific pretraining and highlights how design choices, such as model size and pretraining data, can be tailored to maximize surgical representation learning.

A limitation of our study is that SurgeNetXL is considerably smaller than the LVD datasets used for general-domain DINO pretraining, suggesting that future work should explore substantially larger surgical video collections to further improve SSL performance. At the same time, SSL can be effectively applied to smaller, procedure-specific datasets, which would require less computational resources while still capturing relevant domain-specific representations. This approach could enable targeted pretraining for specialized tasks, maintaining many of the benefits of in-domain SSL without the overhead of large, multi-procedure datasets.

In summary, our study highlights the effectiveness of SSL in surgical computer vision, emphasizing the roles of data domain, model size, and training parameters. These insights provide practical guidance for future work seeking to leverage large self-supervised models in surgical settings.

## 6. Conclusion

In this work, we conducted a systematic study of DINOv1–v3 models for surgical representation learning. Our results show that in-domain pretraining on surgical data enhances performance across semantic segmentation and phase recognition tasks. DINOv2 and DINOv3, particularly with larger architectures, achieve SOTA performance. These findings highlight the value of domain-specific SSL for surgical computer vision and offer practical guidance for future applications.

## Acknowledgments

This research was funded by Stichting Hanarth Fonds, study number 2022-13, and the Dutch Research Council (NWO), study number KICH1.ST03.21.019. It is part of the INTRA-SURGE (INTelligent computeR-Aided Surgical gUidance for Robot-assisted surGEry) project aimed at advancing the future of surgery. Furthermore, we thank SURF (www.surf.nl) for the support in using the Dutch National supercomputer Snellius.

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
