# OpenReview forum: "Towards Effective Surgical Representation Learning with DINO Models"
_MIDL.io/2026/Conference — MIDL 2026 Poster_

### Official Review · Reviewer_fnkB · 2026-01-06

**Confidence:** 4
**Preliminary Rating:** 3
**Final Rating:** 4

**Summary:**

The authors do a systematic validation of Self-supervised training method DINO (v1 to v3) on a large surgical images dataset (SurgeNetXL).
The goal it to validate these methods for real-world medical applications, where labeled data is often sparse, and give recommendations.
Additionally some visualizations are shown of the representations learned.

**Strengths:**

The authors approach this real-world task in a systematic way. The usecase of a large medical dataset with sparse labels is very real, and knowing which methods work well is valuable.
Various combinations of pretraining (using ImageNet vs the SurgeNetXL) datasets are compared, and some detailed analysis with PCA of extracted features is shown to qualitatively show what the features encode.

**Weaknesses:**

- The paper is not very innovative by it's nature, as it's really a comparison of existing methods. The main result is that later version of DINO do indeed perform better, although v3 is not much better than v2. This is not particularly groundbreaking (although the various findings along the way could be interesting).
- Ideally the general benefit of SSL pretraining would also be interesting to see; i.e. comparison to a vanilla network trained from scratch on the limited labeled data.

**Detailed Comments:**

No further comment.

**Justification Of Final Rating:**

The authors do do very diligent experiments; still I think the innovation factor is not high, but if such a 'comparison' paper is seen valuable on this specific task, I can see it work. This is more a call for the chairs; I'll change to weak accept.

**Justification Of The Preliminary Rating:**

The paper gives some relevant insights to applying SSL to a case of a large sparsely labeled dataset; a very real usecase in actual applications. However, the paper also is not exactly advancing any state of the art or adds much novelty by the nature of being an comparison of existing methods. There is some extra analysis that can be interesting to researchers.

**Questions To Address In The Rebuttal:**

Was there a reason why a vanilla network couldn't be trained on the task for comparison (to see how much SSL adds).

---

> ### Author Response · Authors · 2026-01-22
> **Authors’ Rebuttal to Reviewer fnkB for Submission 83**
>
> **1. The paper is not very innovative by it's nature, as it's really a comparison of existing methods. The main result is that later version of DINO do indeed perform better, although v3 is not much better than v2. This is not particularly groundbreaking (although the various findings along the way could be interesting).**
>
> We agree that this paper is not innovative in the sense that it does not introduce a new method, but that is not its goal. Instead, our work provides a systematic, large-scale evaluation of DINOv1, DINOv2, and DINOv3 for surgical representation learning, a domain with unique challenges such as high variability in anatomy, instruments, and procedural context. We benchmark these models on multiple downstream tasks, including semantic segmentation and surgical phase recognition, and demonstrate that in-domain pretraining on a large surgical dataset (SurgeNetXL, 4.7 million frames) consistently improves performance even for models pretrained on massive natural image datasets. Beyond performance comparisons, we analyze the effects of model size, pretraining duration, and frozen versus trainable encoders.
>
> In this revised version, we include additional findings to further enhance the paper, such as comparisons with training models from scratch, evaluations against a broader set of state-of-the-art models, and a more detailed analysis on inference time measurements, and computational pretraining requirements. By providing these empirical insights along with publicly available pretrained models and protocols, our study offers practical guidance for applying state-of-the-art self-supervised learning in surgical computer vision, fills an important knowledge gap, and supports reproducible, high-performing downstream applications.
>
> **2. Ideally the general benefit of SSL pretraining would also be interesting to see; i.e. comparison to a vanilla network trained from scratch on the limited labeled data.**
>
> Thanks for this suggestion. We agree that the general benefit of SSL pretraining was not fully clear in the original version of the paper. To address this, we conducted an additional experiment where all ViT variants were trained from scratch on the downstream tasks. The results, now included in Tables 1 and 2, show that SSL pretraining provides a substantial benefit: in all cases, SSL-based networks consistently outperform their counterparts trained from scratch across datasets, tasks, and ViT variants.

---

### Official Review · Reviewer_cjLu · 2026-01-07

**Confidence:** 4
**Preliminary Rating:** 2
**Final Rating:** 3

**Summary:**

In this study, the authors systematically investigate DINOv1, DINOv2, and DINOv3 for surgical representation learning by pre-training these models on SurgeNetXL and evaluating their transferability on downstream tasks (semantic segmentation, surgical phase recognition). Results show that in-domain pretraining consistently improves performance across all DINO variants.

**Strengths:**

- Source code and models are provided, very good!
- Paper is well structured and easy to follow
- Very detailed evaluation study on 3 DINO models
- Very detailed qualitative analysis, improving the quality of the paper, very nice

**Weaknesses:**

- It is kind of obvious that a model trained on natural images performs worse than a model that is pre-trained on a medical domain which makes the evaluation somewhat difficult to interpret --> SurgeNetXL trained and CholecSeg8k eval are in a similar domain, surgical videos, whereas ImageNet trained and CholecSeg8k eval are completely different domains, doomed to fail (or at least perform way worse than the other one)..
- Further, a major concern is about the effectiveness, in general the results are improved, however we are talking about a range of 2-7% Dice. Whereas on avg it is an improvement about 3-4%. Now training a ViT-b or -l takes a lot of computation, so are the performance increases justifying the computation needed for pre-training, or would someone just stick with the out-of-the-box model without a full training?
- An ablation on computational resources (e.g., runtime per epoch and memory usage) is missing, which is necessary to evaluate whether the method is worth using in practice.
- Why using DINO, if there are Vision Foundation Models which work with Video Materials like SAM-2, where we maybe can leverage the already trained ViT and transfer knowledge into the medical domain with a quick retraining of the head?

**Detailed Comments:**

- Although the paper is in a good shape, my main concern is that It is not clear what the study should achieve/ what its contribution is?

**Justification Of Final Rating:**

The authors addressed most of my comments, although I am still not convinced on the benefit of the method, since training requires immense computational resources (2hrs per epoch and 94GB GPU), while performances are not significantly better.

**Justification Of The Preliminary Rating:**

The reported performance gains are modest (≈3–4% Dice on average) and come at a high computational cost, with no ablation on runtime or memory usage to assess practical viability. Additionally, the evaluation is confounded by models trained on (un)related domains, making it unclear whether the improvements are meaningful or generalizable.

**Questions To Address In The Rebuttal:**

1. Fairness of pre-training comparison:
How should the comparison be interpreted when DINO is pre-trained on a closely related surgical video domain (SurgeNetXL), while ImageNet pre-training is from a completely different domain (compared to CholecSeg8k), making the performance gap largely expected?
2. Cost–benefit of performance gains:
Given that the reported improvements are mostly in the 2–7% Dice range (approx. 3–4% on average), do these gains justify the substantial computational cost of pre-training large ViT-b/l models, compared to using strong off-the-shelf baselines?
3. Missing computational ablations:
Can the authors provide ablations on computational resources (e.g., runtime per epoch, memory usage, and training cost) to assess the practical viability of the method?
4. Choice of pre-training strategy:
What is the motivation for using DINO-based pre-training instead of leveraging existing vision foundation models trained on video data (e.g., SAM-2), which may allow faster and more efficient transfer via lightweight fine-tuning?

---

> ### Author Response · Authors · 2026-01-22
> **Authors’ Rebuttal to Reviewer cjLu for Submission 83**
>
> **1. How should the comparison be interpreted when DINO is pre-trained on a closely related surgical video domain (SurgeNetXL), while ImageNet pre-training is from a completely different domain (compared to CholecSeg8k), making the performance gap largely expected?**
>
> We acknowledge that the domain gap makes it unsurprising that ImageNet-pretrained models underperform compared to SurgeNetXL-pretrained models. However, this comparison is intentional: general computer vision baselines provide a meaningful reference for quantifying the benefits of in-domain surgical pretraining, especially because the default DINOv2-v3 pretrained models are trained on significantly more data (142M and 1689M frames, versus our 4.7M frames). While the performance gap is still not unexpected, our evaluation highlights the magnitude of improvement achievable with domain-specific data across various downstream datasets. It also allows us to analyze how design choices, such as the DINO variant, model size, and other training parameters, affect performance. In other words, this comparison is not intended to suggest that ImageNet pretraining is competitive, but rather to clearly demonstrate the value of surgical-specific pretraining and the improvements it can deliver in different scenarios. We have added a similar explanation to the Discussion section of the revised paper.
>
> **2. Given that the reported improvements are mostly in the 2–7% Dice range (approx. 3–4% on average), do these gains justify the substantial computational cost of pre-training large ViT-b/l models, compared to using strong off-the-shelf baselines?**
>
> To assess this cost benefit trade-off, we added an ablation on computation time (see answer to point 3). These results indicate that the computational cost is significant, while the Dice score improvements are modest, around 2 to 7 percent on average. However, the Dice score gains are consistent across datasets, particularly on smaller segmentation sets, and represent a one-time investment whose benefits extend to multiple downstream tasks, reducing annotation effort and improving reliability. Although the computational cost is non-negligible, it is relatively modest compared to the cost of training lower-performing general computer vision baselines. Moreover, pretraining is typically performed once, after which the resulting models can be reused across many downstream tasks and datasets. To facilitate this, we have made our pretrained models publicly available. For applications requiring self-supervised learning on new datasets outside the scope of our pretraining data, smaller-scale experiments can be performed on datasets smaller than 4.7M images, substantially reducing pretraining cost.
>
> **3. Can the authors provide ablations on computational resources (e.g., runtime per epoch, memory usage, and training cost) to assess the practical viability of the method?**
>
> We agree that computational ablations are important for assessing practical viability and added a new ablation reporting the pretraining cost of DINOv3, presented in Table 4. Specifically, we report memory usage, batch sizes, and runtime per epoch on four NVIDIA H100 GPUs. For ViT-S and ViT-B, training requires 94 GiB per GPU with a runtime of 2.19 hours per epoch (likely because data loading is the bottleneck, not model size), while ViT-L requires the same memory per GPU but 3.36 hours per epoch due to its larger model size. These results provide a clear reference for the computational resources required and highlight the practical feasibility of pretraining DINOv3 on subsets of surgical data. Additionally, we have included an inference efficiency comparison in a new Table 3 to further emphasize the practical feasibility of deploying these models in real-world tasks.
>
> **4. What is the motivation for using DINO-based pre-training instead of leveraging existing vision foundation models trained on video data (e.g., SAM-2), which may allow faster and more efficient transfer via lightweight fine-tuning?**
>
> We appreciate this question and have addressed it by including an additional model, SAM2-UNet, a recently published approach for medical image segmentation that leverages a pretrained SAM 2 model, in our SOTA comparison (Figure 1). While SAM2-UNet achieves strong performance on some benchmarks, our method consistently outperforms it. Foundation models like SAM 2 enable fast transfer through lightweight fine-tuning, but they often lack the fine-grained surgical priors required for precise anatomical and procedural understanding. In contrast, DINO, pretrained on SurgeNetXL, captures domain-specific structures and instruments, resulting in superior segmentation and phase recognition. This highlights that in-domain self-supervised pretraining can surpass off-the-shelf models for tasks that demand high spatial accuracy and clinical reliability.

---

> > ### Author Response · Authors · 2026-02-02
> > **Response to final 'Borderline' rating**
> >
> > We thank Reviewer cjLu for their final assessment. While we appreciate the acknowledgment that we addressed most previous comments, we respectfully disagree with the final justification for a "Borderline" rating. We would like to correct the record regarding the performance gains, which we believe the reviewer has significantly underestimated.
> >
> > **1. Correction of Performance Gains (Dice and HD95) The reviewer’s claim that improvements are only "modest (≈3–4% Dice on average)" is mathematically inaccurate and based on an approximation that does not reflect our full results.**
> >
> > Exact Dice Improvement: We have recalculated the exact average gains for in-domain (SurgeNetXL) pretraining compared to general-domain (ImageNet/LVD) baselines across all DINO versions and architectures based on Table 1. The actual average improvement is 7.0% in Dice score.
> >
> > Substantial HD95 Improvement: The reviewer’s justification relies almost exclusively on Dice scores, overlooking the crucial metric of HD95 (95th percentile Hausdorff distance), which measures boundary alignment and spatial precision. In-domain pretraining achieved an average improvement of 11.2% in HD95.
> >
> > Clinical Significance: In surgical applications, an 11.2% improvement in boundary alignment is not "modest"—it is a substantial gain that directly impacts the safety and reliability of tasks like tool tracking and anatomical margin identification. Dismissing these results as non-significant based on approximated numbers and a lack of investigation into boundary metrics does not accurately represent the contribution of this work.
> >
> > Qualitative Semantic Superiority (PCA): This quantitative improvement is further supported by our qualitative PCA analysis. Features extracted from SurgeNetXL-pretrained models clearly capture the semantic meaning of surgical images, representing anatomical structures as distinct, coherent clusters. In contrast, models pretrained on natural images (LVD-1689M) produce noisier and less semantically meaningful representations for the same surgical scenes.
> >
> > **2. The Efficiency of One-Time Pretraining We believe the concern regarding the "immense computational resources" (2–3 hours per epoch) is misplaced in the context of foundation model development.**
> >
> > Amortized Cost: Pretraining is a one-time investment. Once the SurgeNetXL weights are trained and released, the broader research community can achieve these 7–11% gains without repeating the pretraining process. This is also why we provided all model weights on Github and Huggingface.
> >
> > Frozen Encoder Utility: Our study shows that a frozen encoder (which requires zero gradient updates to the backbone) performs nearly as well as a fully trainable one. This makes the resulting models highly accessible for practical, low-resource downstream fine-tuning at other institutions. In the end, our one-time computational investment might lead to a lower overall computational footprint, because fine-tuning can be done about 10 times quicker on a large variety of surgical AI tasks and datasets.
> >
> > **3. State-of-the-Art (SOTA) Superiority**
> >
> > Finally, we must emphasize that our SurgeNetXL-pretrained DINO models do not just improve over ImageNet or standard DINO pretraining; they outperform existing SOTA surgical foundation models like SAM2-UNet and EndoFM. Our models consistently maintain the top rank in stability across all evaluated datasets and tasks.
> >
> > **Conclusion**
> >
> > The claim that the method lacks benefit is based on an incomplete and mathematically approximated view of the results. When considering the 7.0% Dice gain, the 11.2% HD95 gain, the one-time nature of pretraining, and the SOTA rankings, the practical and scientific value of this study is clear. We hope the Program Committee takes these specific, recalculated figures into account.

---

### Official Review · Reviewer_MGZP · 2026-01-08

**Confidence:** 5
**Preliminary Rating:** 4
**Final Rating:** 5

**Summary:**

The authors investigate DINOv1, DINOv2, and DINOv3 for surgical representation learning. They pretrain DINO models with a large-scale surgical dataset named SurgeNetXL, and evaluate the models on downstream tasks, including semantic segmentation and surgical phase recognition. Experiments across multiple datasets demonstrate that the proposed approach achieves state‑of‑the‑art performance.

**Strengths:**

(1) The authors utilized multiple datasets to evaluate their methods. They experiment with different versions and model sizes of DINO to provide a more comprehensive analysis.

(2) The authors include an ablation study comparing frozen and trainable encoders during fine‑tuning, which demonstrates the high quality of DINO features.

**Weaknesses:**

(1) The stability analysis is conducted only on segmentation datasets. The authors should also compare their DINO models against state‑of‑the‑art approaches on surgical phase recognition datasets to fully assess representation quality across tasks.

(2) Given that the proposed representations are intended for downstream real‑time applications, inference time analysis should be conducted.

(3) The paper would benefit from comparisons against a broader set of state‑of‑the‑art methods, including those that do not rely on large‑scale in‑domain pre‑training but still achieve strong performance on small datasets. For each dataset, ranking the proposed method alongside these baselines would clarify its relative standing and more convincingly demonstrate its advantages.

(4) Please check all references and update the reference if necessary.

**Detailed Comments:**

For weakness point (3), the authors could address this by adding a separate table, or by explicitly highlighting which models use pretraining within the existing tables.

(4) Please check all references and update the reference if necessary.

For example:

The paper titled “Unsupervised representation learning by predicting image rotations” has already published.

**Justification Of Final Rating:**

I would like to thank the authors for their response. The newly added experiments and the inference‑time analysis substantially improve the overall quality of the paper. While I agree with the other reviewers that the novelty is somewhat limited, I find the work to be a solid and valuable application. Taking all factors into account, I am raising my rating to a strong accept.

**Justification Of The Preliminary Rating:**

This work is primarily an application‑focused paper from my point of view. The downstream task analysis remains incomplete. It also lacks a real‑time inference speed evaluation. I recommend a weak accept.

**Questions To Address In The Rebuttal:**

Please prioritize addressing weaknesses (1), (2), and (3)

---

> ### Author Response · Authors · 2026-01-22
> **Authors’ Rebuttal to Reviewer MGZP for Submission 83**
>
> **1. The stability analysis is conducted only on segmentation datasets. The authors should also compare their DINO models against state of the art approaches on surgical phase recognition datasets to fully assess representation quality across tasks.**
>
> We thank the reviewer for the suggestion. We agree and have added this comparison to our paper in Figure 1, which shows that, on surgical phase recognition, our model also achieves superior performance, demonstrating the strength and generalizability of our learned representations across tasks.
>
> **2. Given that the proposed representations are intended for downstream real time applications, inference time analysis should be conducted.**
>
> We agree that this is a valuable addition and have included an inference time analysis in a new Table 3. The inference experiments show that all DINO variants support real-time deployment for surgical video tasks. Smaller ViT models achieve the highest FPS, surpassing 200 FPS for segmentation, while even the largest models remain above 90 FPS. Phase recognition runs slightly slower but still satisfies real-time requirements, as frame rates around 30 FPS are sufficient for most practical scenarios. Our analysis shows that there is a clear trade-off between model size and speed, which becomes particularly important when deploying on devices with limited computational resources.
>
> **3. The paper would benefit from comparisons against a broader set of state of the art methods, including those that do not rely on large scale in domain pre training but still achieve strong performance on small datasets. For each dataset, ranking the proposed method alongside these baselines would clarify its relative standing and more convincingly demonstrate its advantages. For weakness point (3), the authors could address this by adding a separate table, or by explicitly highlighting which models use pretraining within the existing tables.**
>
> We have added two additional state-of-the-art methods that do not rely on large-scale in-domain pretraining but still achieve strong performance on small datasets. Specifically, we included a SegFormer model and SAM2-UNet in our ranking stability analysis (Figure 1). In this figure, we also explicitly highlighted which models use self-supervised pretraining. These newly added models perform well compared to other self-supervised learning approaches, but their performance remains lower than that of our proposed model.
>
> **4. Please check all references and update the reference if necessary.**
>
> Thanks for spotting this mistake. We have updated all references to their latest published versions, including the paper “Unsupervised Representation Learning by Predicting Image Rotations” and a few other papers.

---

### Author Rebuttal · Authors · 2026-01-22

**Rebuttal:**

We thank the reviewers for their insightful comments, which have led to numerous improvements in the manuscript. We have carefully revised the paper to address all reviewer suggestions. All revisions and additions are highlighted in green in the attached renewed manuscript. Our responses to each of the reviewers’ comments can be found in the official comments underneath each review.

**Supporting Material:**

/attachment/a9198dc1361bc51c9aeba17a5a00d73db82602ad.pdf

---

### Meta-Review · Area_Chair_FBRj · 2026-02-09

**Recommendation:** Accept (Poster)
**Confidence:** 5

**Metareview:**

This paper received **strong accept**, **weak accept**, and **borderline**.

The final decision reflects the reviewers’ overall evaluations. The authors are encouraged to carefully address the remaining concerns and incorporate key clarifications from the rebuttal into the final manuscript.

The primary outstanding concern is that the training procedure requires substantial computational resources (approximately 2 hours per epoch and 94 GB of GPU memory), while the performance gains over existing methods are not clearly significant.

---

### Decision · Program_Chairs · 2026-02-13

Accept (Poster)